# Effects of a Mixture of Ivy Leaf Extract and *Coptidis rhizome* on Patients with Chronic Bronchitis and Bronchiectasis

**DOI:** 10.3390/ijerph18084024

**Published:** 2021-04-12

**Authors:** Goohyeon Hong, Yu-Il Kim, Seoung Ju Park, Sung Yong Lee, Jin Woo Kim, Seong Hoon Yoon, Keu Sung Lee, Min Kwang Byun, Hak-Ryul Kim, Jaeho Chung

**Affiliations:** 1Department of Internal Medicine, Division of Pulmonary and Critical Care Medicine, Dankook University Hospital, Dankook University College of Medicine, Cheonan 31116, Korea; hkh0519@hanmail.net; 2Department of Internal Medicine, Division of Pulmonology, Chonnam National University Hospital, Chonnam 61469, Korea; kyionly@chonnam.ac.kr; 3Department of Internal Medicine, Division of Pulmonology, Allergy and Critical Care Medicine, Jeonbuk National University Medical School, Cheonbuk 54907, Korea; sjp@jbnu.ac.kr; 4Department of Internal Medicine, Division of Pulmonology, Allergy and Critical Care Medicine, Korea University Guro Hospital, Seoul 08308, Korea; pusarang@gmail.com; 5Department of Internal Medicine, Division of Pulmonology, College of Medicine, The Catholic University, Uijeongbu 11765, Korea; medkjw@catholic.ac.kr; 6Department of Internal Medicine, Division of Pulmonology, Pusan National University Yangsan Hospital, Pusan 49241, Korea; drysh79@gmail.com; 7Department of Internal Medicine, Division of Pulmonary and Critical Care Medicine, Ajou University Hospital, Suwon 16499, Korea; plator@aumc.ac.kr; 8Department of Internal Medicine, Division of Respiratory Medicine, Gangnam Severance Hospital, Seoul 06273, Korea; littmann@yuhs.ac; 9Department of Internal Medicine, Division of Respiratory Medicine, Wonkwang University Hospital, Iksan 54538, Korea; kshryj@wku.ac.kr; 10Department of Internal Medicine, Division of Respiratory Medicine, Catholic Kwandong University International St. Mary’s Hospital, Incheon 22711, Korea

**Keywords:** chronic bronchitis, bronchiectasis, hederacoside C, berberine, mucolytic agent

## Abstract

Background: Hederacoside C from ivy leaf dry extracts (HH) and berberine from *Coptidis rhizome* dry extracts (CR) can be combined (HHCR) as a herbal product. Previous studies have demonstrated that HHCR has antitussive and expectorant effects in animal models of respiratory disease. However, the therapeutic effects of HHCR on respiratory diseases in humans have not been well-studied. Therefore, we aimed to clarify the effectiveness of HHCR in patients with chronic bronchitis and bronchiectasis. Methods: This was a multicenter (10 university teaching hospitals), open-label, prospective, single-arm, observational study. Consecutive patients with chronic bronchitis and bronchiectasis were included. Patients were orally treated with HHCR daily for 12 weeks. St. George’s Respiratory Questionnaire (SGRQ) scores and bronchitis severity scores (BSS) were measured at baseline and at the end of the 12-week study. Results: In total, 376 patients were enrolled, of which 304 were finally included in the study, including 236 males and 68 females with a median age of 69 years (range: 37–88 years). After 12 weeks of HHCR treatment, there was a significant improvement in SGRQ score (baseline, 32.52 ± 16.93 vs. end of study, 29.08 ± 15.16; *p* < 0.0001) and a significant reduction in BSS (baseline, 7.16 ± 2.63 vs. end of study, 4.72 ± 2.45; *p* < 0.0001). During the study, 14 patients concomitantly used an inhaled corticosteroid and 83 patients used an inhaled bronchodilator. HHCR also had significant positive effects on these patients in terms of SGRQ score and BSS. No serious adverse drug reactions occurred during HHCR treatment. Conclusions: treatment with HHCR improved the SGRQ score and BSS in patients with chronic bronchitis and bronchiectasis. HHCR may be a new therapeutic option for chronic bronchitis and bronchiectasis. Large-scale, randomized, double-blind, placebo-controlled clinical trials are warranted.

## 1. Introduction

Bronchiectasis is a chronic respiratory disorder associated with poor quality of life (QOL) and frequent exacerbations. There are some international guidelines on treatment of bronchiectasis, including The European Respiratory Society (ERS) and British Thoracic Society (BTS) guidelines. Chronic bronchitis may degrade lung function and increase the rate of exacerbation, resulting in clinical consequences such as aggravate health-related QOL and increase the possibility of mortality [1]. The severity of symptoms in patients with bronchiectasis and chronic bronchitis is related to chronic mucus hypersecretion and mucus plugging. Impairment of mucociliary transport is induced by the impact of structural changes in airway cellular architecture, airway dehydration, and excess mucus volume and viscosity. More than 70% of bronchiectasis patients expectorate highly variable volumes of sputum daily. Treatment aims to prevent mucus stasis and plugging, airflow obstruction, and progressive lung damage [2].

Large quantities of *Coptidis rhizome* (CR) are consumed in Asian countries, including China, Japan, Malaysia, Singapore, and India, but very little is used in European countries [3]. The therapeutic characteristics of berberine, the main ingredient of CR, are the suppression of inflammation by inhibiting the lipopolysaccharide (LPS)-stimulated pro-inflammatory cytokines, such as interleukin-6 and LPS-mediated nuclear factor (NF)-kB activation [4]. Kim et al. [5]. reported that berberine suppresses the expression of the *MUC5AC* gene, which is primarily involved in mucin production in chronic inflammatory airway diseases, and some studies have shown that berberine exhibits bronchodilatory and antimicrobial activities [6]. Hederacoside C (HH), the aglycone part of ivy leaf extract, has long been used to treat respiratory diseases by blocking inflammatory mediators such as bradykinin to exert anti-inflammatory effects [7].

HHCR is a herbal product consisting of HH from ivy leaf dry extracts and berberine from CR dry extracts. HHCR is approved in the Republic of Korea for the relief of cough and sputum formation due to acute respiratory infection and inflammation. The research on CR and HH has mostly focused on molecular mechanisms using animal studies [5,6,8,9]. Although the mechanism of action of HHCR is known, details of its therapeutic effects on chronic bronchitis and bronchiectasis are not well-known. This study investigated the effectiveness of oral HHCR in patients with chronic bronchitis and bronchiectasis.

## 2. Materials and Methods

### 2.1. Study Design

This 12-week, open-label, multicenter, prospective, single-arm, observational study was conducted at 10 university teaching hospitals in the Republic of Korea. The first patient was enrolled on 25 September 2017, and the last patient was examined on 28 June 2018. The study protocol was designed by the investigator steering committee and presented to all study locations thereafter. All investigators were trained before the trial to ensure reliable data quality, with special emphasis on understanding the protocol. The study was conducted in accordance with the Declaration of Helsinki, and Good Clinical Practice Guidelines. The protocol was approved by the institutional review board at each study center, and all patients gave written informed consent before participating in any study procedures.

Patients were seen at the initial visit, at week 4 after enrollment, and at week 12 of the study; any unscheduled visits were recorded. Patients were treated orally with fixed dose 15 mg of HHCR three times daily (Synatura^®^; Ahn-Gook Pharmaceutical Co., LTD., Seoul, Korea) for 12 weeks. At every visit, adherence to the study regimen was assessed. The study flowchart is shown in Figure 1. No follow-up study was planned.

Salbutamol, a short-acting β2-agonist, was administered as needed to relieve symptoms. Maintenance treatment for chronic obstructive pulmonary disease (COPD) that started prior to the study, including the use of short-acting bronchodilators, long-acting bronchodilators such as long-acting β2-agonists and long-acting muscarinic antagonists, inhaled corticosteroids (ICS), and methylxanthines such as theophylline and doxofylline, was permitted to continue during the study period, whereas the use of mucoactive drugs, antitussives, antibiotics, and systemic corticosteroids was prohibited to avoid any potential confounding effects on clinical endpoints.

### 2.2. Patient Participation

Patients aged ≥35 years, regardless of whether they were smokers or non-smokers, were eligible if they had been diagnosed with chronic bronchitis based on the Global Initiative for Chronic Obstructive Lung Disease (GOLD) guidelines [10] or bronchiectasis. The definition of chronic bronchitis is chronic cough and sputum production for at least 3 months per year for 2 consecutive years. Bronchiectasis was diagnosed using a computed tomography scan and previously recorded diagnostic criteria findings.

Exclusion criteria were: acute exacerbation of chronic bronchitis, continuous treatment with systemic corticosteroids, active infection due to *Mycobacterium tuberculosis* (except for patients with small, old tuberculous lesions), pneumonia, history of active peptic ulcer disease or intestinal malabsorption, congestive heart failure categorized as class 2 or higher according to the New York Heart Association, severe neurological disease, severely impaired hepatic or renal function, immunocompromised status, suspected or known hypersensitivity to the study product or any of its excipients, rare hereditary problem of fructose intolerance, pregnancy or active lactation status, poor reliability (e.g., history of alcohol or drug abuse, mental disorder), and poor compliance.

### 2.3. Efficacy Outcomes and General Measures

The primary endpoint was the change in St. George’s Respiratory Questionnaire (SGRQ) score between the baseline and week 12. The SGRQ is disease-specific instrument designed to assess of quality of life. It is consists of 50 items divided into three components: symptom component, activity component, and impact component. Each score of components and total score are calculated. The SGRQ is displayed with a score ranging from 0 to 100, where 0 indicates the best quality of life related to health, and the higher score means a poorer quality of life.

The secondary endpoint was the change in Bronchitis Severity Score (BSS) between the baseline and week 12. The BSS was developed to measure an appropriate outcome of bronchitis [11]. The scale comprises the five most important features of bronchitis, i.e., coughing, sputum production (expectoration), rales/rhonchi (auscultation), chest pain while coughing, and dyspnea. This instrument combines objective and subjective items, because the assessment is based on the investigator’s clinical evaluation in conjunction with subjective feedback from the patient. Each constituent of the BSS is assessed by the investigator using a five-point verbal rating scale ranging from 0 to 4 (0, absent; 1, mild; 2, moderate; 3, severe; 4, very severe). The total score is the sum of the five ratings with a maximum of 20 points. Since its introduction in 1996, the BSS has been successfully used in many clinical studies as a main outcome measure for patients suffering from bronchitis [12]. An additional efficacy endpoint was the association between variation in SGRQ scores and variation in BSS.

Patients were first seen at a screening visit during which inclusion/exclusion criteria, medical and medication history, and the status of exacerbations were checked, and the results of a physical examination, chest radiograph, electrocardiography (ECG), laboratory tests, and pregnancy test as well as the SGRQ score and BSS were recorded. Patients who met the inclusion criteria were asked to sign the informed consent form. After the start of the study, patients returned to the investigation center at weeks 4 and 12. At each visit, the following information was recorded: physical examination results, pregnancy test results, concomitant medications (including ICS, and short- and long-acting bronchodilators), occurrence of spontaneous adverse events (AEs), and level of compliance. At week 12, the SGRQ and BSS were assessed again.

Safety assessments included the recording of AEs, laboratory tests, blood pressure measurements, and 12-lead ECG. All adverse events or concurrent illnesses during the study were documented in the clinical record form. Major adverse cardiovascular events (MACE; a composite of total cardiovascular death, non-fatal myocardial infarction, and non-fatal stroke) were evaluated and classified by an independent, blinded adjudication committee. AEs were evaluated and presented descriptively. An AE was defined as any untoward medical occurrence in a patient who was administered the study drug, regardless of whether it was considered to be related to the study drug. An AE was considered serious if it was either immediately life threatening or resulted in persistent or significant disability, prolonged hospitalization, or death.

### 2.4. Statistical Analysis

All data are presented as medians or the means ± standard deviations (SDs) for continuous variables and as numbers with percentages in parentheses for categorical variables. The primary positive outcome in the study would be an improvement in the SGRQ score and BSS after 12 weeks of HHCR treatment. Changes from baseline to the endpoints were analyzed using the paired *t*-test. All statistical analyses were performed using SAS for Windows software (9.4; SAS Institute Inc., Cary, NC, USA). A value of *p* < 0.05 was considered to denote significance.

## 3. Results

### 3.1. Study Population and Baseline Characteristics

Of 377 eligible patients with chronic bronchitis or bronchiectasis who were recruited, one patient was excluded during the screening period due to withdrawn consent. Of the 72 (19.2%) patients who discontinued treatment, 27 (7.2%) were lost to follow-up, 15 (4%) had protocol violations, 14 (3.7%) withdrew consent, 5 (1.3%) had AEs, 5 (1.3%) were non-compliant, and 6 (1.6%) discontinued for other reasons.

Patient demographics and baseline characteristics are summarized in Table 1. The median age of the patients was 69 years (range, 37–88 years) at inclusion, and male patients accounted for 77.6% (236 patients) of the patient group. The mean body mass index (BMI) of the whole group was 22.83 ± 3.33. Pharmacologic treatments for chronic bronchitis and bronchiectasis, taken occasionally or routinely during the last three months prior to enrolment, are listed in Table 2.

### 3.2. Primary Endpoint

The median total SGRQ score was 29.96 (range, 4.07–96.16) and the mean was 32.52 ± 16.93 (SD) at baseline. At week 12, a median score of 25.64 (range, 3.86–89.35) and a mean score of 29.08 ± 15.16 were reported. Patients treated with HHCR exhibited a statistically significant and clinically relevant improvement from baseline in terms of SGRQ total score, with median reduction in score of −2.37 and a mean reduction of −3.44 ± 10.97 (*p* < 0.0001) after 12 weeks of treatment (Figure 2). Of the SGRQ domains, statistically significant and clinically relevant improvements were seen in the symptoms (median score reduction, −2.94; mean score reduction, −5.01 ± 12.21; *p* <0.0001) and impact domains (median score reduction, −2.28; mean score reduction, −4.37 ± 13.33; *p* < 0.0001) after 12 weeks. The activity domain score also improved (mean score reduction, −0.97 ± 14.76; *p* = 0.0971), but the difference was not statistically significant (Figure 2).

### 3.3. Secondary Endpoint

The total median and mean BSS at baseline were 7.0 (range, 2.0–15.0) and 7.16 ± 2.63, respectively, and the total median and mean scores at week 12 were 4.50 (range, 0.0–13.0) and 4.72 ± 2.45, respectively. Patients treated with HHCR exhibited a statistically significant and clinically relevant improvement from baseline in terms of total BSS, with a median score reduction of −2.0 and a mean score reduction of −2.45 ± 2.36 (*p* < 0.0001) after 12 weeks of treatment (Figure 3). Of the BSS components, statistically significant and clinically relevant improvements were seen after 12 weeks in terms of scores related to coughing (median score reduction, −1.0; mean score reduction, −0.53 ± 0.77; *p* < 0.0001), sputum production (median score reduction, −1.0; mean score reduction, −0.77 ± 0.88; *p* < 0.0001), auscultation (median score reduction, 0.0; mean score reduction, −0.49 ± 0.76; *p* < 0.0001), chest pain while coughing (median score reduction, 0.0; mean score reduction, −0.20 ± 0.56; *p* < 0.0001), and dyspnea (median score reduction, 0.0; mean score reduction, −0.46 ± 0.80; *p* < 0.0001) (Figure 3). The correlation coefficient for the relationship between the SGRQ score and BSS was 0.3766 (*p* < 0.0001). A scatter plot of the variation in BSSs vs. variation in SGRQ scores is shown in Figure 4.

### 3.4. Changes in SGRQ Score and BSS in Patients Concomitantly Treated with a Bronchodilator

In total, 83 patients were treated concomitantly with a bronchodilator during the 12-week study period. For these patients, the total median SGRQ score was 35.81 (range, 5.57–96.16) and the mean score was 37.66 ± 18.54 at baseline. At week 12, a median score of 29.86 (range, 7.99–89.35) and a mean score of 34.51 ± 17.74 were reported. Patients treated with HHCR exhibited a statistically significant and clinically relevant improvement from baseline in terms of SGRQ total score, with a median score reduction of −1.71 and a mean score reduction of −3.15 ± 13.31 (*p* = 0.0115) after 12 weeks of treatment. Statistically significant and clinically relevant improvements after 12 weeks were seen in the symptoms (mean score reduction, −5.19 ± 15.83; *p* = 0.0049) and impact (mean score reduction, −3.84 ± 15.21; *p* = 0.014) domain scores. The activity domain score improved (mean score reduction, −0.80 ± 17.38; *p* = 0.4788), but the improvement was not statistically significant (Figure 5).

The median and mean total BSS at baseline were 7.0 (range, 2.0–13.0) and 6.88 ± 2.75, respectively, and the median and mean total scores at week 12 were 4.0 (range, 0.0–12.0) and 4.55 ± 2.35, respectively. The change in total BSS from baseline to week 12 reflected a statistically significant and clinically relevant improvement, with a median score reduction of −2.0 and a mean score reduction of −2.33 ± 2.66 (*p* < 0.0001). Of the BSS components, statistically significant and clinically relevant improvements were seen after 12 weeks in terms of scores related to coughing (median score reduction, −1.0; mean score reduction, −0.48 ± 0.94; *p* < 0.0001), sputum production (median score reduction, −1.0; mean score reduction, −0.66 ± 0.91; *p* < 0.0001), auscultation (median score reduction, 0.0; mean score reduction, −0.37 ± 0.79; *p* < 0.0001), chest pain while coughing (median score reduction, 0.0; mean score reduction, −0.12 ± 0.50; *p* = 0.0514), and dyspnea (median score reduction, −1.0; mean score reduction, −0.69 ± 0.85; *p* < 0.0001) (Figure 5).

### 3.5. Safety and Tolerability

Based on pooled analysis of safety data, 30 AEs were reported by 28 patients (9.21%) from the treatment population during the 12-week study. Most AEs were mild in severity and generally well-tolerated. The most frequent AEs were respiratory symptoms, including coughing, dyspnea, rhinorrhea, and sore throat, followed by gastrointestinal symptoms such as nausea, gastric irritation, and heartburn. Neurologic symptoms such as insomnia and myalgia were also reported. Of the 30 AEs, only four were determined by the investigators to be related to the study drug. A summary of reported AEs is shown in Table 3.

In all, three patients (0.99%) reported three serious AEs, including two cases of dyspnea and one increase in prostate specific antigen. However, two of the serious AEs were determined by the investigators to be unrelated to the study drug. Only one patient (0.3%) discontinued HHCR treatment because of a serious AE (dyspnea). All the AEs improved spontaneously during treatment with HHRC; therefore, these AEs were not causally related to HHCR treatment. During the 12-week observation period, no MACE or fatal outcomes were reported. No abnormal findings were observed in chest radiographs, ECG outputs, or laboratory tests, including tests for full blood count, blood chemistry, and urinalysis during the study period.

## 4. Discussion

Bronchiectasis is a chronic respiratory disease, radiologically characterized by abnormal and permanent expansion of the bronchi, and clinical syndromes such as sputum production, cough, and bronchial infection [13]. Sputum production, coughing, and breathlessness are the most frequent symptoms [14]. Treatment is primarily based on the principles of preventing exacerbations, suppressing acute and chronic bronchial infection, reducing symptoms, improving mucociliary clearance and QOL, and preventing disease progression [14]. Exacerbation of bronchiectasis is the main objective for treatment since it is related to airway increase and systemic inflammation [15] as well as progressive lung damage [16,17]. In addition, more severe and frequent exacerbations are associated with worse QOL, daily symptoms [18], lung function decline [19], mortality, and higher health-care costs. Despite current treatment approaches, about half of patients with bronchiectasis suffer more than two exacerbations per year, and about 33% patients require at least one hospitalization per year [20].

Chronic bronchitis result from goblet cells producing and secreting excessive mucus. By luminal obstruction of small airways, epithelial remodeling, and alteration of airway surface tension, it contributes to aggravating airflow obstruction and finally make the airway vulnerable to collapse. Despite the clinical sequelae, there is only limited information on the pathophysiology of chronic bronchitis and goblet cell hyperproliferation of COPD, and there are insufficient alternatives for treatment [1]. Impairment of mucociliary clearance is a representative pathologic feature of bronchiectasis and chronic bronchitis, and it results in various respiratory symptoms, including coughing and sputum production [2].

There has been a persistent effort to use traditional natural products as respiratory medications to alleviate coughing and sputum production for thousands of years. HHCR is a natural medicine reported to suppress pulmonary inflammation, and it has been shown to have expectorant and antitussive effects in animals [9]. QOL impairment in chronic bronchitis and bronchiectasis is equivalent in terms of SGRQ scores to severe COPD, idiopathic pulmonary fibrosis, and other disabling respiratory diseases [21,22]. In this study, we focused on the symptomatic improvement of patients with chronic bronchitis and bronchiectasis after treatment with HHCR by evaluating the effects of HHCR on SGRQ score and BSS.

The results of our study indicate that HHCR is able to reduce the respiratory symptoms of patients with chronic bronchitis and bronchiectasis. Treatment with HHCR improved both SGRQ score and BSS. Moreover, subgroup analysis of the changes in SGRQ score and BSS among patients concomitantly treated with a bronchodilator showed that the SGRQ score and BSS improved in this patient group compared with the group with no concomitant bronchodilator treatment. Previous study has suggested that HHCR is a potential useful therapeutic option for respiratory diseases [9]. However, there have been few studies on the effects of HHCR in humans. This is the first study to elucidate the possible benefits of HHCR for the treatment of patients with chronic bronchitis and bronchiectasis.

HH is a triterpene saponin and an active ingredient derived from ivy leaf extracts. It exhibits biological properties such as expectorant and bronchospasmolytic effects, and hence has been therapeutically applied for the treatment of chronic inflammatory bronchial conditions and productive coughs [7,23,24]. CR comprises various alkaloid constituents, such as berberine, coptisine, palmatine, and epiberberine; among them, berberine is the most potent. Berberine have bronchodilating and antimicrobial action and may suppress the mucin in sputum [6]. These actions are accomplished by a mechanism that inhibits the expression of IL-1β-induced MUC5AC gene in human airway epithelial cells through ERK and p38MAPK. This indicates that berberine has antitussive and expectorant effects.

Chronic bronchitis and bronchiectasis with acute exacerbations are difficult to treat, resulting in high morbidity and mortality. Unfortunately, no foundational treatment agent has been identified. Some mucolytics and expectorants such as N-acetylcysteine, erdostein, and ambroxol are used for reducing mucus secretion and promoting mucus clearance, but there have been concerns about their effectiveness, limitations, and side effects [25]. Our study suggests that a course of oral HHCR might have symptomatic and clinical advantages in patients with chronic bronchitis and bronchiectasis.

Nevertheless, the study has some limitations. First, this was an open-label, single-arm, observational trial, and treatment with a bronchodilator and ICS was permitted. This open design might have biased the results, and the use of concomitant treatment is a possible confounding factor. However, there were no significant differences in the use of bronchodilator treatments from baseline to week 12 in all enrolled patients, and HHCR treatment was uniformly administrated to all patients in the study. Additionally, regarding the bronchodilator, out study’s bronchodilator uses were not specialized as short-acting or long-acting. Therefore, the difference in effect between each type of bronchodilator is limited. Despite this limitation, this pilot clinical trial is the first one performed in patients. Further large-scale placebo-controlled studies are warranted. Second, objective parameters of airway inflammation such as cytokines, chemokines, and sputum inflammatory cells were not estimated. However, the focus of this study was to determine an individual’s health-related QOL change using validated and widely accepted instruments such as the SGRQ and BSS. Third, our study’s design investigated 12 weeks only, so further further safety evaluations for extended periods and patient outcomes needed (e.g., exacerbation rate after treatment completed). However, clinical improvements and no serious AEs after 12 weeks of treatment could be considered reasonable treatment goals for chronic bronchitis and bronchiectasis. Finally, we did not evaluate correlations between HHCR and lung function/clinical parameters. Further large-scale clinical trials are needed to elucidate these correlations.

## 5. Conclusions

Despite some limitations, the results of this observational, real-world trial revealed that patients with chronic bronchitis and bronchiectasis experienced considerable and clinically relevant improvements in health-related QOL, with improvements in the total BSS score and all five domain scores as well as the SGRQ total score and all three domain scores after treatment with HHCR. The changes in SGRQ score and BSS among patients concomitantly treated with a bronchodilator showed that the SGRQ score and BSS improved in this patient group compared with the group with no concomitant bronchodilator treatment. The potential attenuation of symptom progression associated with chronic bronchitis and bronchiectasis reported in this study suggests that HHCR may be a reasonable therapeutic agent for further prospective placebo-controlled trials.

## Figures and Tables

**Figure 1 ijerph-18-04024-f001:**
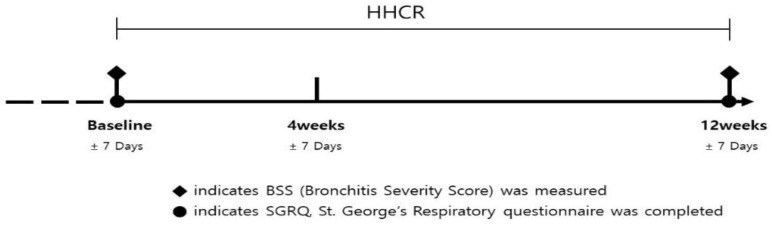
Study design.

**Figure 2 ijerph-18-04024-f002:**
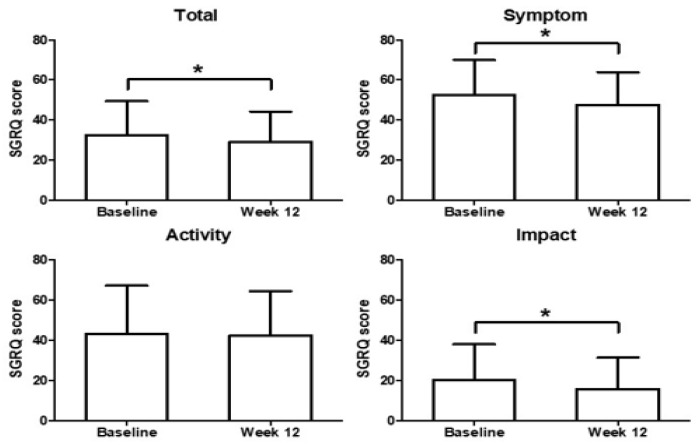
Change from baseline in St. George’s Respiratory Questionnaire (SGRQ) total scores, activity domain, impact domain, and symptom domain. *p*-value < 0.0001 (Cf. Activity *p*-value: 0.0971) Wilcoxon’s signed-rank test. *: Statistically significant difference.

**Figure 3 ijerph-18-04024-f003:**
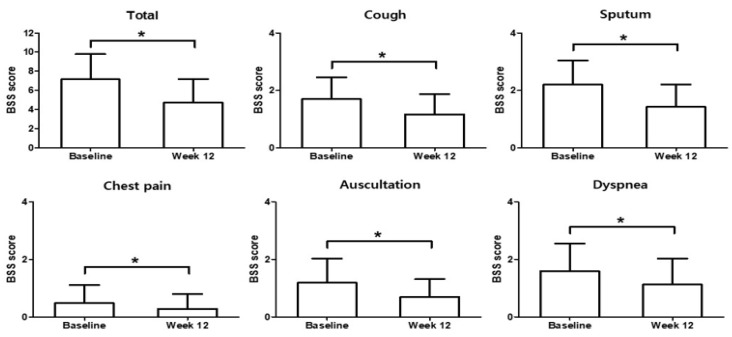
Change from baseline in bronchitis severity scores (BSS) total scores, cough, sputum production, chest pain, auscultation, and dyspnea. *p*-value < 0.0001 (Cf. Activity *p*-value: 0.0971) Wilcoxon’s signed-rank test. *: Statistically significant difference.

**Figure 4 ijerph-18-04024-f004:**
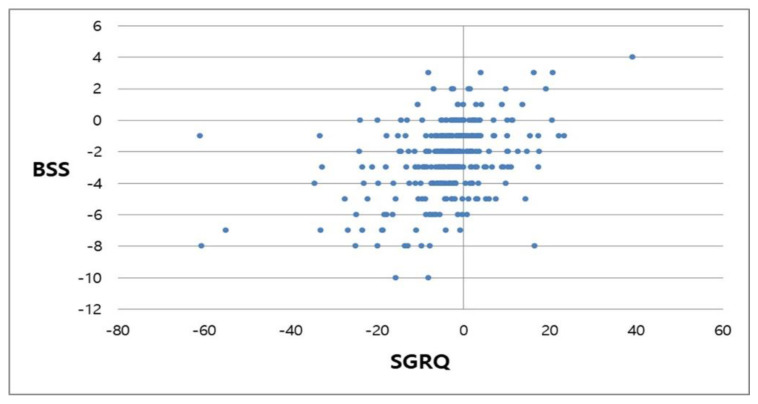
Scatter plot of variation between SGRQ (St. George’s Respiratory Questionnaire) and BSS (Bronchitis Severity Score).

**Figure 5 ijerph-18-04024-f005:**
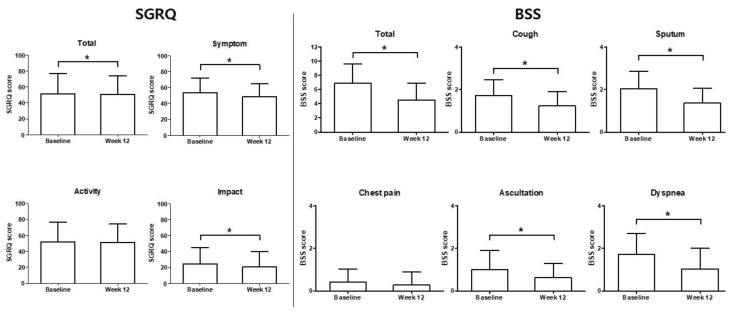
Change of SGRQ and BSS in patients who had concomitant treatment with bronchodilator *p*-value < 0.0001 (Cf. ‘’Chest pain’’ *p*-value: 0.0514) Wilcoxon’s signed-rank test. *: Statistically significant difference.

**Table 1 ijerph-18-04024-t001:** Baseline characteristics of the study population.

	Total (*n* = 304)
Sex, *n* (%)	
Male	236 (77.63)
Female	68 (22.37)
Age (year)	
Mean ± SD	67.88 ± 8.94
Median	69.00
Min, Max	37.00, 88.00
Height (cm)	
Mean ± SD	163.27 ± 7.70
Median	164.00
Weight (kg)	
Mean ± SD	60.93 ± 10.31
Median	60.35
Min, Max	37.00, 90.00
BMI (kg/m^2^)	
Mean ± SD	22.83 ± 3.33
Median	22.75
Min, Max	13.68, 31.60

*n* (%), number of patients (percentage of patients); SD, standard deviation; Min, minimum; Max, maximum; BMI, body mass index.

**Table 2 ijerph-18-04024-t002:** Pharmacologic treatment for chronic bronchitis and bronchiectasis at entry.

Drug Name	Total (*n* = 304)
Use of any concomitant drug (*n* (%))	
ICS alone	
Fluticasone furoate	3 (0.99)
Budesonide	1 (0.33)
Ciclesonide	10 (3.29)
ICS-LABA	
Beclometasone/Formoterol	7 (2.30)
Fluticasone/Formoterol	6 (1.97)
Fluticasone/Salmeterol	6 (1.97)
Budesonide/Formoterol	20 (6.58)
Fluticasone/Vilanterol	16 (5.26)
LABA	
Tulobuterol	1 (0.33)
LAMA	
Aclidinium bromide	9 (2.96)
Umeclidinium bromide	1 (0.33)
Tiotropium bromide	33 (10.86)
LABA-LAMA	
Olodaterol/Tiotropium bromide	27 (8.88)
Indacaterol/Glycopyrronium bromide	23 (7.57)
Vilanterol/Umeclidinium bromide	23 (7.57)
Mucolytics and antioxidant	
Erdosteine	40 (13.16)
Carbocysteine	1 (0.33)
Ambroxol	3 (0.99)
Bromhexine	3 (0.99)
Acetylcystein	16 (5.26)
Others	
Doxofylline	33 (10.86)
Theophylline	6 (1.97)
Roflumilast	8 (2.63)
Pranlukast	26 (8.55)
Montelukast	15 (4.93)
Theobromine	11 (3.62)

*n* (%), number of patients (percentage of patients); ICS, inhaled corticosteroid; LABA, long-acting beta-agonist; LAMA, long-acting muscarinic antagonist.

**Table 3 ijerph-18-04024-t003:** Summary of reported adverse events (AEs).

	Total (*n* = 304)
Any AEs	28 (9.21%)
Respiratory	16
Gastrointestinal	7
Nervous system	3
Others	2
Any SAE	3
Respiratory	2
Others	1
Any ADR	4
Respiratory	1
Gastrointestinal	2
Nervous system	1
Any SADR	0
Hospitalizations	0
MACE/Fatal AEs	0

AE, adverse event; *n*, number; SAE, serious adverse event; ADR, adverse drug reaction; SADR, severe adverse drug reaction; MACE, major adverse cardiovascular events.

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
