# Peer review of "Effects of a Mixture of Ivy Leaf Extract and Coptidis rhizome on Patients with Chronic Bronchitis and Bronchiectasis"

_ijerph, 2021, doi:10.3390/ijerph18084024_

Round 1

Reviewer 1 Report

At the introduction, authors wrote about bronchiectasis that "there are no international guidelines on treatment", they must review better, there are european, british, spanish guidelines about bronchiectasis.

At methods and results, some comments and suggestions:

  • Patient participantion: sample calculation?
  • You don't describe how may participants had bronchiectasis or chronic bronchitis
  • The participants excluded (73), were they different in baselines characteristics from those who were finally analyzed (304)?
  • Why were excluded 5 participants with AE at the beginning, and after you describe (Table 3) a summary of reported AEs of the rest of participants.
  • Why dont record spirometry? bronchiectasis severity (BSI, FACED)?
  • Did you record baseline symptoms? (cough, expectoration, volume of expectoration) ¿chronic respiratory infections? ¿Other comorbidities?

Nowadays, patients with bronchiectasis could be characterized better in the context of observational studies, so that clinical trials can be designed better looking for a greater impact in this population. This is somethingI cannot find in this study, the design must be improved.   

Author Response

Reviewer #1 (Comments to the Author):

  1. At the introduction, authors wrote about bronchiectasis that "there are no international guidelines on treatment", they must review better, there are european, british, spanish guidelines about bronchiectasis.

Response : Thank you for your comment. It exists that International Guideline including The European Respiratory Society (ERS) and British Thoracic Society (BTS) guidelines as the reviewer’s comment. We corrected as reviewer's comment. 

At methods and results, some comments and suggestions:

  1. Patient participantion: sample calculation?

Response : Since this is an non-interventional observational study, no statistical formula was not applied for sample calculation. the actual number of cases that can be registered during the study period was calculated.

  1. You don't describe how many participants had bronchiectasis or chronic bronchitis

Response : Thank you for your comment. In this study, there were some patients overlapping bronchiectasis and chronic bronchitis characteristics. So we didn't exactly count two categories, respectively. Since this study was conducted with two indications, it would be possible to confirm more accurate results for each indication if further sub-analysis was conducted by classifying the chronic bronchitis group and the bronchiectasis group. Based on this result, additional RCT will be conducted to derive more specific results.

  1. The participants excluded (73), were they different in baselines characteristics from those who were finally analyzed (304)?

Response : The baselines of all enroll patients were collected, but dropout patients were not included in the statistics. We didn’t analyze the characteristics of dropout patients.

  1. Why were excluded 5 participants with AE at the beginning, and after you describe (Table 3) a summary of reported AEs of the rest of participants.

Response : Five dropout patients were excluded in order to analyze the risk and benefit of patients who performed all studies according to the protocol.

  1. Why don’t record spirometry? bronchiectasis severity (BSI, FACED)?

Response : This study is a non-interventional observational pilot study, so we have minimized test items. We have focused evaluation for relief of patients’ symptoms and as a questionnaire tool to measure the quality of life in patients with chronic lung disease, SGRQ is considered to be the most valid and reliable in many ways. From the viewpoint of symptom evaluation, only one BSS was selected. We have already described this limitation in the discussion.

  1. Did you record baseline symptoms? (cough, expectoration, volume of expectoration) chronic respiratory infections? Other comorbidities?

Response : Five symptoms (cough, sputum, chest pain, auscultation, and dyspnea) with baseline corresponding to BSS and comorbidity of all patient was collected, but were not included in the context of the manuscript. (comorbidity result attached)

  1. Nowadays, patients with bronchiectasis could be characterized better in the context of observational studies, so that clinical trials can be designed better looking for a greater impact in this population. This is something cannot find in this study, the design must be improved.

Response : Thank you for your valuable comment. This is the pilot study. As a pilot study, the design may not be smooth, but we intend to proceed with a large scale study in the future with a improved design that shows the effect on patients with bronchiectasis and bronchitis based on this study.   

Reviewer 2 Report

How was the SGRQ score and BSS defined? Please describe them in more details, since the entire manuscript is based on these.

In line 15-16 in the introduction section, the authors claim: “The therapeutic characteristics of berberine, the main ingredient of CR, are the suppression of anti-inflammatory protein expression and blocking of nuclear factor-kB signaling” citing reference number 4.

This is not appropriate, in that Choi et al describe in their paper the inhibitory effects of Coptis chinensis on inflammation and not the suppression of anti-inflammatory protein. Please verify. 

Although the study evaluates the effects of HHCR for only 12 weeks, what are the long-term beneficial effects?

according to the authors, is it possible to administer HHCR for the whole life of the patient on a daily basis, since he is suffering from chronic diseases?

Furthermore, by what molecular mechanism could HHCR induce an improvement in the symptoms of such patients?

the studies cited in the introduction refer to the possible mechanisms of action of HH and CR separately;

so what is the advantage of using their combination?

Author Response

  1. How was the SGRQ score and BSS defined? Please describe them in more details, since the entire manuscript is based on these.

Response : Thank you for your comment. We have added the more details of The St. George's Respiratory Questionnaire (SGRQ) in the Materials and Methods section. The SGRQ is a disease specific instrument designed to assess of quality of life. It consists of 50 items divided into three components: symptom component, activity component, and impact component. Each score of components and total score are calculated. The SGRQ is displayed with a score ranging from 0 to 100, where 0 indicates the best quality of life related to health, and the higher score means a poorer the quality of life.

BSS was scored in 5 levels (0: none, 1: mild, 2: moderate, 3: severe, 4: very severe) for the main symptoms of bronchitis in a total of 5 items (cough, sputum, blisters, chest pain, dyspnea). It is divided, and a decrease in the BSS score means improvement in symptoms. We already described about BSS in the Materials and Methods section (2.3 Efficacy outcomes and general measures) and cite the reference. (Reference No. 11)

  1. In line 15-16 in the introduction section, the authors claim: “The therapeutic characteristics of berberine, the main ingredient of CR, are the suppression of anti-inflammatory protein expression and blocking of nuclear factor-kB signaling” citing reference number 4.

This is not appropriate, in that Choi et al describe in their paper the inhibitory effects of Coptis chinensis on inflammation and not the suppression of anti-inflammatory protein. Please verify. 

Response : Thank you for your comment. This is our mistake. We have corrected as your comment. The therapeutic characteristics of berberine, the main ingredient of CR, are the supression of inflammation by inhibiting the LPS-stimulated pro-inflammatory cytokines(such as interleukin-6) and LPS-mediated nuclear factor(NF)-κB activation.

  1. Although the study evaluates the effects of HHCR for only 12 weeks, what are the long-term beneficial effects?

Response : According to our study results, adverse effects were found in about 9% of patients, mostly mild and well tolerated AE. In addition, only four of them are considered to have a causal relationship with drugs, so even if a long-term study is conducted later, it is considered to be effective in chronic lung disease without significant side effects.

  1. According to the authors, is it possible to administer HHCR for the whole life of the patient on a daily basis, since he is suffering from chronic diseases?

Response : In patients with chronic disease, if they have respiratory symptoms, continued administration of HHCR will be helpful for relief of respiratory symptoms. HHCR, which has fewer side effects, may have safety benefits.

  1. Furthermore, by what molecular mechanism could HHCR induce an improvement in the symptoms of such patients?

Response : We have mentioned molecular mechanism of HH and CR in 2nd paragraph of introduction and 5th paragraph of discussion. Further study is needed to demonstrate molecular mechanism of HHCR combination is whether different from that of separate ingredient.

  1. The studies cited in the introduction refer to the possible mechanisms of action of HH and CR separately;

so what is the advantage of using their combination?

Response : It is thought to have a synergistic effect in alleviating symptoms of bronchitis. According to the Phase 3 clinical trial (data not published, studied by Ahn-Gook Pharmaceutical Co., LTD., Seoul, Republic of Korea, funding Co.) in 235 patients, HHCR showed more than twice the effect of single drug on chronic bronchitis. In recognition of this effectiveness, it was approved by the Ministry of Food and Drug Safety.

Reviewer 3 Report

The manuscript "Effects of a mixture of ivy leaf extract and Coptidis rhizome on patients with chronic bronchitis and bronchiectasis" is interesting. But some major corrections are required to improve the manuscript quality.

Comments:

  • The plagiarism check result using Turnitin Software shows a more than 30% similarity index in the text (excluding materials and methods, and References). 
  • Explain the route of administration of HHCR?
  • How you normalized the dose of HHCR in different patients with different clinical conditions?
  • In conclusion, explain clearly that the treatment with HHCR alone or in combination with current therapy is effective.
  • How can you differentiate the effects of short-acting bronchodilators+ HHCR and long-acting bronchodilators + HHCR? I would recommend including additional data set or results differentiating categorized short-acting and long-acting bronchodilators with HHCR (SGRQ score and BSS).

Author Response

The manuscript "Effects of a mixture of ivy leaf extract and Coptidis rhizome on patients with chronic bronchitis and bronchiectasis" is interesting. But some major corrections are required to improve the manuscript quality.

Comments:

  1. The plagiarism check result using Turnitin Software shows a more than 30% similarity index in the text (excluding materials and methods, and References). 

Response : So far, many articles have been published on bronchiectasis and chronic bronchitis. Since we have cited the contents of various researches such as review articles, editorials and clinical trials for the bronchiectasis and chronic bronchitis, there may be some unintentional similarities. However, all the contents of other researches have been accurately cited and all references corresponding to them are accurately described in this manuscript.

  1. Explain the route of administration of HHCR?

Response : HHCR is administered orally as a suspension syrup. We already described the route of administration in the Material and Methods (2.1 Study design)

  1. How you normalized the dose of HHCR in different patients with different clinical conditions?

Response : HHCR is a commercially available drug, and has been approved for administration of 15ml three times a day for adults in South Korea. Therefore, dose of HHCR was not normalized for each patients.

  1. In conclusion, explain clearly that the treatment with HHCR alone or in combination with current therapy is effective.

Response : We have already described that the results of subgroup analysis comparing SQRQ and BSS according to the use of bronchodilator demonstrated that HHCR and bronchodilator may be more effective when used together in the disscussion. We added it back to the conclusion as your comment.

  1. How can you differentiate the effects of short-acting bronchodilators+ HHCR and long-acting bronchodilators + HHCR? I would recommend including additional data set or results differentiating categorized short-acting and long-acting bronchodilators with HHCR (SGRQ score and BSS).

Response : Thank you for your valuable comment. This study was a pilot study, and the purpose was to check whether any effect when bronchodilator is used with HHCR without distinction of bronchodilator. It will be used as the basis for further research. To overcome limitation of this study, well designed additional large-scale clinical trial that distinguishes short and long-acting bronchodilators will be conducted.

Round 2

Reviewer 1 Report

No comments

Author Response

done

Reviewer 3 Report

The revised version of the manuscript "Effects of a mixture of ivy leaf extract and Coptidis rhizome on patients with chronic bronchitis and bronchiectasis" is interesting. 

Major comments:-

I strongly recommend authors check plagiarism rules and avoid copy and pasting in the introduction and discussion part.

If HHCR dose normalization was not performed in different patients, I recommend authors explain this in the manuscript with adequate references (15ml three times a day for adults in South Korea). 

Author Response

eviewer #3
1. Please check plagiarism rules and avoid copy and pasting in the introduction and discussion part.

We have edited as you recommended. By copykiller plagiarism check, it has been decrease to 11% from 29%. The remaining concordances are the terms essential to explaining our research.

2. If HHCR dose normalization was not performed in different patients, authors, please explain this in the manuscript with adequate references (15ml three times a day for adults in South Korea).

Thank you for your suggestion. We have added a reference to the approval information provided by the Ministry of Food and Drug Safety, which is in charge of drug approval. (Please check ‘synatura_approval information.pdf’)
Reference: Ministry of Food and Drug Safety_synatura®
https://nedrug.mfds.go.kr/pbp/CCBBB01/getItemDetail?itemSeq=201101306
The contents of the attached pdf file are translated as follows.

Original translation
용법용량

연령에 따라 아래의 용량으로 1일 3회 경구투여
2 – 6세 : 1회 5ml
7 – 14세 : 1회 10ml
15세 이상 : 1회 15ml
Dosage

Orally administered 3 times a day at the following doses depending on age

2-6 years old: 5ml per time
7-14 years old: 10ml per time
≥15 years old : 15ml per time

[Revised part of Manuscript]
1) 1.Introduction 3rd paragraph line 2
2) 2.Materials and Methods-2.1 study design 2nd paragraph line 4
Reference (26) has been added
3) 2.Materials and Methods-2.1 study design 2nd paragraph line 3
Patients were treated orally with fixed dose 15 mg of HHCR three times daily

3. Recommendation of additional figure or table differentiating categorized short-acting and long-acting bronchodilators with HHCR (SGRQ score and BSS) from the available data set
Thanks for the thoughtful recommendation. As you said, we agree that adding an additional figure or table is a way to explain the significance of the study in more detail. However, as a pilot study, this study focused on whether it is effective when used with HHCR regardless of the type of bronchodilator when analyzing the overall data. Therefore, it is difficult to analyze additional data. We will reflect what you recommended in a future large-scale study. Not only for our future study but also for the researchers who will conduct additional research through our literature, we have added the limitation of not differentiating the type of bronchodilator in the discussion.

[Revised part of Manuscript]
4. Discussion 7th paragraph line 6
Additionally, regarding the bronchodilator, used in this study was not classified as short-acting and long-acting, but was analyzed together. Therefore, the difference in effect between each type of bronchodilator is limited.
